# Live cell imaging of single genomic loci with quantum dot-labeled TALEs

Yingxin Ma[1,3,*], Mingxiu Wang[1,*], Wei Li[1], Zhiping Zhang[1], Xiaowei Zhang[1], Tianwei Tan[3], Xian-En Zhang[2] & Zongqiang Cui[1]

Single genomic loci are often related to specific cellular functions, genetic diseases, or pathogenic infections. Visualization of single genomic loci in live human cells is currently of great interest, yet it remains challenging. Here, we describe a strategy for live cell imaging of single genomic loci by combining transcription activator-like effectors (TALEs) with a quantum dot labelling technique. We design and select a pair of TALEs that specifically target HIV-1 proviral DNA sequences, and use bioorthogonal ligation reactions to label them with different colour quantum dots (QDs). These QD-labelled TALEs are able to enter the cell nucleus to provide fluorescent signals to identify single gene loci. Based on the co-localization of the pair of different coloured QD-labelled TALEs, we determine and map single-copy HIV-1 provirus loci in human chromosomes in live host cells.

[1] State Key Laboratory of Virology, Wuhan Institute of Virology, Chinese Academy of Sciences, Wuhan 430071, China. [2] CAS Center for Biological Macromolecules, National Key Laboratory of Biomacromolecules, Institute of Biophysics, Chinese Academy of Sciences, Beijing 100101, China. [3] Beijing Key Laboratory of Bioprocess, College of Life Science and Technology, Beijing University of Chemical Technology, Beijing 100029, China. * These authors contributed equally to this work. Correspondence and requests for materials should be addressed to Z.C. (email: czq@wh.iov.cn) or to X.-E.Z. (email: x.zhang@wh.iov.cn).

Single genomic loci in human chromosomes are often related to specific cellular functions, human genetic diseases and pathogen infections. Live visualization of single genomic loci would help to elucidate the spatiotemporal positions, dynamics and function of different loci and offer understanding of genome architecture and regulation[1]. However, imaging of single genomic loci in live cells is currently challenging. DNA-fluorescent *in situ* hybridization can be used for chromosomal visualization, but must be performed on chemically fixed cells and is thus unsuitable for live cell imaging[2–4]. Recently, several approaches have been developed to image endogenous genomic loci in live cells using fluorescent zinc-finger proteins, transcription activator-like effectors (TALEs), or a modified CRISPR/Cas9 system[5–7]. However, these approaches usually allow the visualization of repetitive genomic sequences. Single-copy genes are difficult to detect using current techniques because of their limited sensitivity, despite zinc-finger proteins, TALEs and the CRISPR/Cas9 systems all being directed to a single-copy sequence[8].

In this work, by combining the sequence-specific recognition of TALEs with the optical superiority of quantum dots (QDs), in applications such as single-particle sensitivity, we propose a strategy for the visualization of single genomic loci in live cells. We demonstrate the utility of this strategy through the imaging of single-copy HIV-1 provirus loci in live cells. It is recognized that the integration of HIV-1 proviral DNAs into human chromosomes represents a major obstacle to eradicating the virus, which makes the AIDS a difficult disease to cure[9–11]. There are often only a few copies of proviral DNA within each host cell[12], rendering the visualization of single HIV-1 provirus loci in live human cells impossible with current techniques.

Our strategy is shown in Fig. 1. We intend to use a pair of QD-TALEs with different coloured fluorescent tags to label and image the single genomic loci of the HIV-1 provirus within the nucleus of a live cell. The TALEs are labelled with different colour QDs within a single live cell via two alternative bioorthogonal ligation reactions. One of the TALEs is fused to a short LplA acceptor peptide (LAP) and is labelled with a tetrazine-conjugated red QD (QD625) via the Diels–Alder cycloaddition[13]. The second TALE is fused to an AP tag and biotinylated and labelled by streptavidin-conjugated green QDs (QD525)[14]. These QDs-TALEs are carried into the cell nucleus via the nuclear localization sequence of the TALEs, bind to the target HIV-1 proviral DNA sequences and provide a fluorescence signal for each single QD-TALE within a cell nucleus. Colocalization microscopy can then be employed to examine the bound QD-TALEs, potentially allowing single-copy HIV-1 provirus loci to be visualized and determined in live cells.

## Results

**TALEs for HIV-1 proviral DNA targeting.** To test our hypothesis, we first designed and selected TALEs specifically for the HIV-1 proviral DNA sequence. We selected the HIV-1 *U3* and *R* regions of the 5′ long terminal repeat (LTR) as target sequences because of their relatively high conservation in different strains[12] (Fig. 2a). Six pairs of TALEs were designed to target the HIV-1 5′ LTR sequence, including R1, R2, R3, N1, N2 and N3 (Fig. 2b, Supplementary Fig. 1), and their binding activity was subsequently characterized by use of yeast-based reporter assay[15]. Five pairs of the designed TALEs were able to bind their target sequences (Fig. 2c). The pair of N2 sequences with the highest binding activity was selected for subsequent experiments.

**Diels–Alder cycloaddition reactions to generate QD-TALEs.** Two different bioorthogonal ligation reactions were used to label the pair of N2 TALEs (N2-L and N2-R) with different colour QDs. First, we used Diels–Alder cycloaddition chemistry to label the TALE N2-L with QDs in live cells[13]. Within our system, this ligation chemistry occurred between the *trans*-cyclooctene TCO2-decorated TALE and tetrazine Tz1-conjugated QD625. The TALE N2-L fused to a 13-amino acid LAP was expressed in live cells, and the LAP tag was site specifically decorated with *trans*-cyclooctene TCO2, which is catalysed by a mutant of the *Escherichia coli* enzyme lipoic acid ligase (LplA) expressed within the same cell. The tetrazine Tz1-conjugated QD625 was delivered into the cell and ligated onto the TALE-LAP fusion protein via a chemoselective derivatization process to yield a highly sensitive, fluoregenically labelled TALE N2-L (Fig. 3a).

The organic molecule TCO2 was chemically synthesized and identified through nuclear magnetic resonance spectroscopy analysis, as seen in Fig. 3b. The tetrazine Tz1-conjugated QD625 was also verified by nucleic acid gel electrophoresis (Fig. 3c).

The Diels–Alder cycloaddition-based labelling method was carried out in a human monocytic cell line, U1 with HIV-1 provirus integrated into the cell's chromosomes. As shown in Fig. 4a, red QD625 fluorescence signals were observed in the nucleus of the U1 cells following the Diels–Alder cycloaddition labelling system. When any single step of the Diels–Alder cycloaddition chemistry was absent, this QD labelling result could not be achieved. For example, when Tz1-QDs were delivered into the cell without the trans-cyclooctene TCO2-decorated TALE, the QDs signals localized to the cytosol and were absent from the nucleus (Fig. 4b). Similarly, omitting the LplA enzyme from the TALE-LAP molecule prevented Tz1-QDs from being linked with the TALE and being transported into the nuclei,

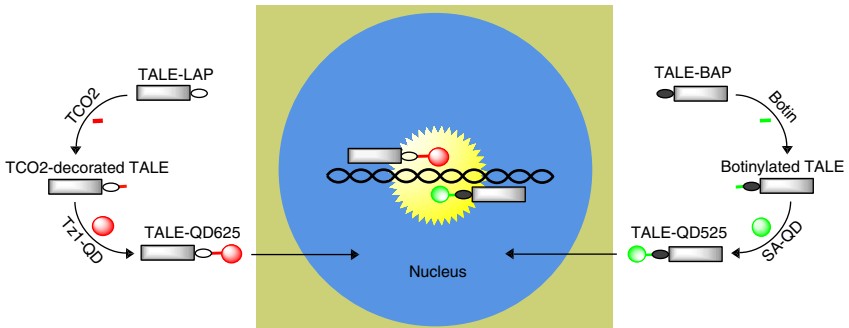

**Figure 1 | Schematic of imaging genomic loci with quantum dot-labelled TALEs.** Within the cytosol of a live cell, the TALEs fused with a short LplA acceptor peptide (LAP) is decorated with *trans*-cyclooctene and subsequently labelled with tetrazine-conjugated red QD (QD625) via Diels–Alder cycloaddition chemistry. The TALE fused with an AP tag is biotinylated and labelled with streptavidin-conjugated green QDs (QD525). The two QD-TALEs bind to the target HIV-1 proviral DNA sequences, and their fluorescence co-localization demonstrates single-copy HIV-1 provirus loci in human chromosomes.

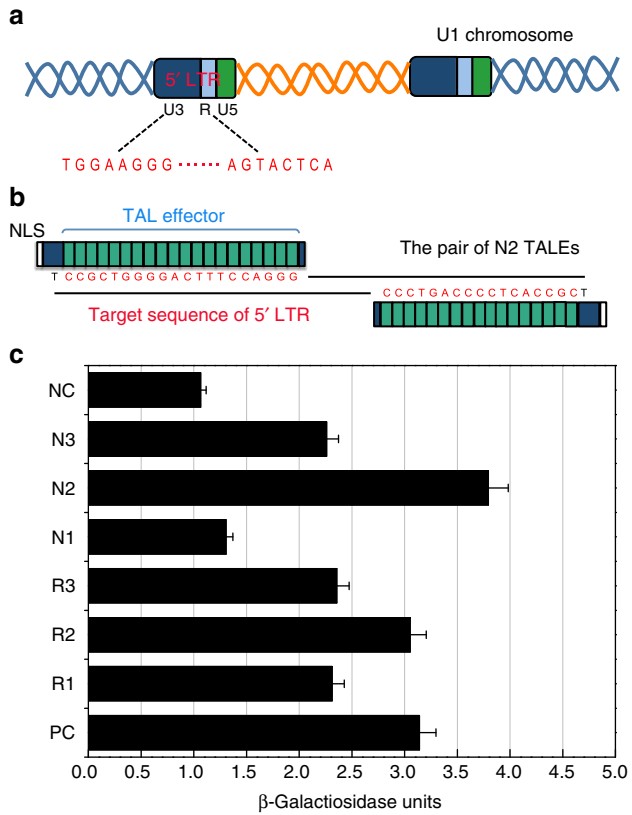

**Figure 2 | Design and selection of TALEs for HIV-1 proviral DNA targeting.** (**a**) Schematic of HIV-1 proviral genome illustrating the distribution of *U3*, *R* and *U5* regions in 5′-LTR. (**b**) Schema depicting the pair of N2 TALEs and their binding sites on the HIV-1 proviral genome. The constructed TALEs contain the central DNA-binding domain with 19.5 and 16.5 repeat units (green boxes), respectively, and nuclear localization sequence at the N-terminus. The target sequences are highlighted in red. (**c**) Activity of six custom TALE pairs targeting diverse sequences of LTR in a reporter-based yeast assay. Activity was measured in a yeast-based assay in which cleavage and recombination reconstitutes a functional *lacZ* gene. Activity was normalized to a PC positive control, and NC indicated mock control. Error bars denote s.d.; $n = 3$.

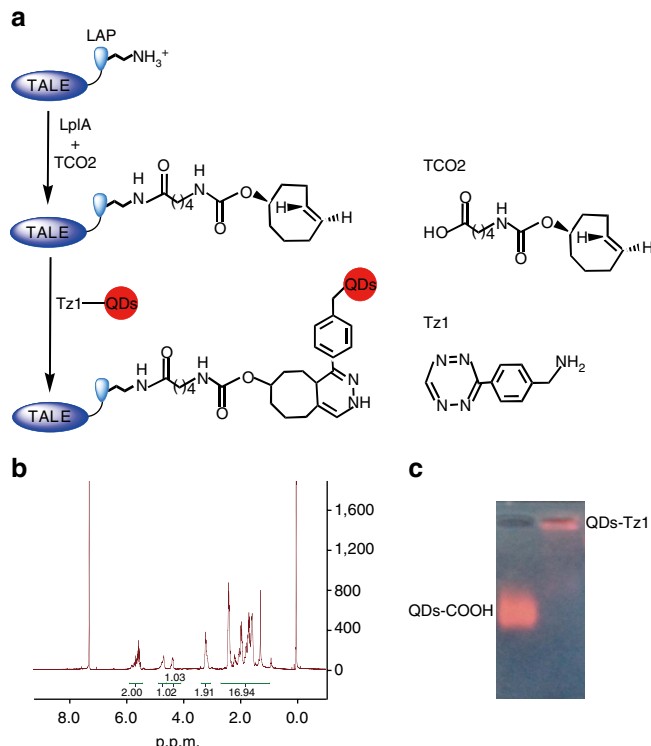

**Figure 3 | The Diels–Alder cycloaddition reaction to generate QD-TALEs.** (**a**) Schematic of fluorescence labelling of TALE N2-L with red QDs. First, *trans*-cyclooctene TCO2 is ligated onto the LAP-tag fused to the TALE N2-L. Second, the TCO2-decorated TALE is conjugated with the tetrazine-QDs (Tz1-QDs) via the Diels–Alder cycloaddition. Molecular formula of TCO2 and Tz1 are shown on the right side. (**b**) $^1$H nuclear magnetic resonance spectrum of ∼5 mg of TCO2 in CDCl$_3$ and its species assignments. (**c**) Agarose gel electrophoresis of native QDs and conjugated with tetrazine.

implying that only QDs linked with the TALE binders can be carried into the nuclei. The nuclear localization of the TALE protein was confirmed using a GFP-TALE fusion protein that localized to the cell nuclei (Fig. 4c,d). Taken together these results demonstrate that the QDs were conjugated with TALEs via the designed Diels–Alder cycloaddition labelling system and can subsequently enter the cell nucleus. To our knowledge, this is also the first time that QDs have been used to specifically label a protein, via the Diels–Alder cycloaddition, within live cells.

To achieve a relatively high labelling efficiency and a low level of imaging background, we applied an optimal experimental condition using two transfection steps. U1 cells were transfected with plasmid encoding LplA and cultured overnight. Then, the cells were transfected with QDs and plasmids encoding TALEs, and imaged under microscopy following further incubation in medium containing TCO2. Under this condition, 56.4% of the QD625s were re-localized to the nucleus by TALEs and the Diels–Alder cycloaddition system (Fig. 4e). When the QDs were not conjugated with TALE, the percentage of QDs being relocalized to the nucleus was 0 (Fig. 4e).

**Biotin–streptavidin labelling to generate QD-TALEs.** In the cell nuclei, the fluorescent TALE N2-L-QD625s could be detected in U1 cells by confocal microscope. However, the presence of excess QD-TALEs within the nucleus prevents the specific localization of HIV-1 proviral DNA loci within the cell genome. To overcome this problem and allow specific localization of the target loci, a second QD-TALE was employed with a different colour fluorescein to enable colocalization.

The TALE N2-R was conjugated to a green-coloured QD525 using a biotin–streptavidin, interaction-based labelling method[14]. A 15-amino-acid peptide (AP tag) was fused to TALE N2-R and specifically biotinylated by biotin ligase (BirA) in live cells. Further addition of streptavidin-conjugated QD525 (SA-QD525) allows the labelling of TALE N2-R with QD525, via the natural interaction of biotin and streptavidin (Supplementary Fig. 2).

Such a labelling method has been previously applied to labelling cell surface proteins and enveloped viruses with QDs[14]. Here, this labelling method was successfully used to label TALEs with QDs in live U1 cells, and green QD525 fluorescence signals were observed in the nucleus of U1 cells (Fig. 5a). Control reactions in which only SA-QD525 was transfected into the cell, showed that the QDs signals were localized within the cytosol but absent from the nucleus (Fig. 5b). Equally, reactions that included no BirA enzyme or biotin moiety prevented the AP tag from being biotinylated, thus blocking ligation to the SA-QD525, showed no nuclear signal. Under our labelling condition, 53.2% of the QD525s were re-localized to

the nucleus by TALEs and the biotin–streptavidin conjugation system (Fig. 5c).

**QD-TALEs imaging allows visualization of single gene loci.** To determine the specific localization of HIV-1 proviral loci within U1 cells, we employed a combination of the two different bioorthogonal labelling methods for concurrent labelling of a pair of TALEs with two different coloured QDs. TALE N2-L was labelled *in vivo* with red QD625 through the Diels–Alder cycloaddition chemistry, and TALE N2-R was labelled *in vivo*

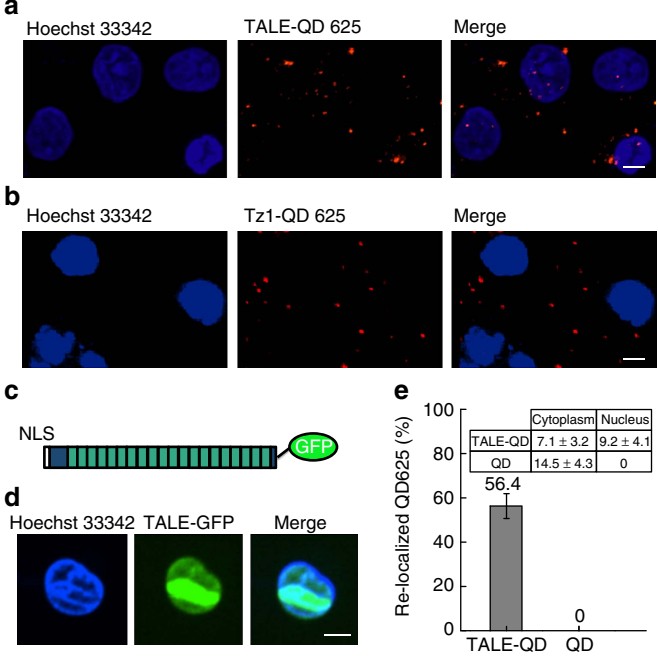

**Figure 4 | Labeling TALEs with QD625 via the Diels–Alder cycloaddition in live cells.** (**a**) Confocal images showing TALE N2-L labelled with QD625 in live cells. (**b**) Confocal images showing Tz1-QD625 transfected into a live U1 cell. (**c**) Diagram of the TALE-GFP-fusion protein. (**d**) Confocal images showing TALE-GFP transiently expressed in living cells (scale bar, 5 μm). (**e**) Percentage of QD625s (2,000 QD625s in 125 cells) with or without attachment to TALE re-localized to the nucleus. Inset: count and statistics of QD625s with or without attachment to TALE in the cytoplasm and nucleus. Error bars denote s.d.; *n* = 125.

with green QD525 through the biotin–streptavidin-based labelling method. Both N2-L-QD625s and N2-R-QD525s were transported into the cell nuclei to bind to the target HIV-1 proviral DNAs. Colocalization of the N2-L-QD625 and N2-R-QD525 was observed in the cell nuclei of U1 cells, showing that the two probes specifically target HIV-1 proviral DNA (Fig. 6a, Supplementary Fig. 3). Colocalization by confocal microscopy demonstrated that a single HIV-1 provirus locus could be determined in live U1 cells by this method. Live cell imaging allows tracking of HIV-1 provirus in real time (Supplementary Movie 1). Our results also showed that the two different QD labelling methods are fully biocompatible and orthogonal to one another. There is no crosstalk between the two labelling systems because LplA specifically ligates the TCO2 onto the LAP tag and BirA specifically catalyses the biotinylation of the AP tag (Fig. 6b, Supplementary Fig. 4). An ultra-sensitive immuno-DNA fluorescence *in situ* hybridization (FISH) experiment[16,17], was also performed to validate that the TALE-QDs co-staining sites were real TALE-targeted HIV DNA loci rather than random aggregations. As shown in Fig. 6c, Supplementary Fig. 5, co-localization of TALE-QDs with the HIV DNA FISH sites stained by Alexa Fluor 647-tagged HIV-specific DNA probes could be observed. A negative control experiment using non-targeting TALEs (N1) was also performed. As shown in Fig. 6d, the QD625s and QD525s were still be translocated into the nucleus when linked to the non-targeting TALEs, but they failed to show a colocalization signal representing an HIV-specific locus. Statistical analyses showed that when the two different QD labelling methods were used in the same cells, 51.5% of the QD625s and 51.7% of the QD525s were re-localized to the nucleus by the two TALE labelling systems respectively, and 21.3% of the cells exhibited the HIV-specific colocalization signals (Fig. 6e).

**Using QD-TALEs to map HIV-1 provirus loci in live U1 cells.** Previous reports suggest that two copies of the HIV-1 proviral DNA are integrated within chromosome X and chromosome 2 respectively in U1 cells[12]. Using our novel labelling method, two dots representing the colocalization of N2-L-QD625 and N2-R-QD525 could be detected in individual U1 cells. Genome labelling was analysed using three-dimensional confocal microscopy to determine whether the colocalized dots were within the same focus plane and therefore truly colocalized. As shown in Fig. 7a, optical sections of the cell nucleus of a U1 cell were acquired along the z axis by confocal microscopy.

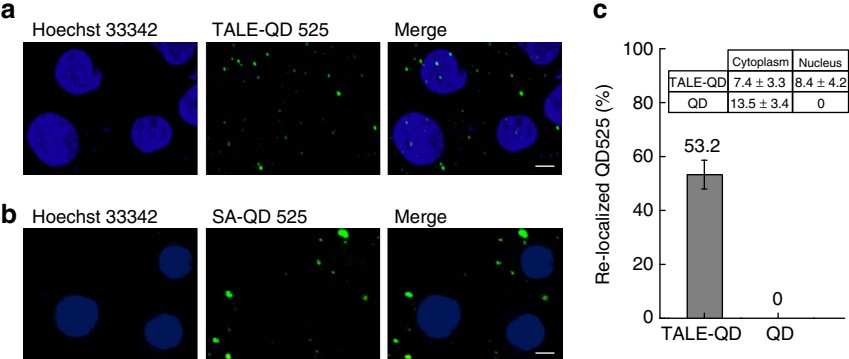

**Figure 5 | Biotin–streptavidin labelling to generate QD-TALEs in live cells.** (**a**) Confocal images showing TALE N2-R labelled with QD525 in live U1 cells. (**b**) Confocal images showing SA-QD525 transfected into a live cell. Scale bar, 5 μm. (**c**) Percentage of QD525s (2,000 QD525s in 129 cells) with or without attachment to TALE re-localized to the nucleus. Inset: Count and statistics of QD525 with or without attachment to TALE in the cytoplasm and nucleus. Error bars denote s.d.; *n* = 129.

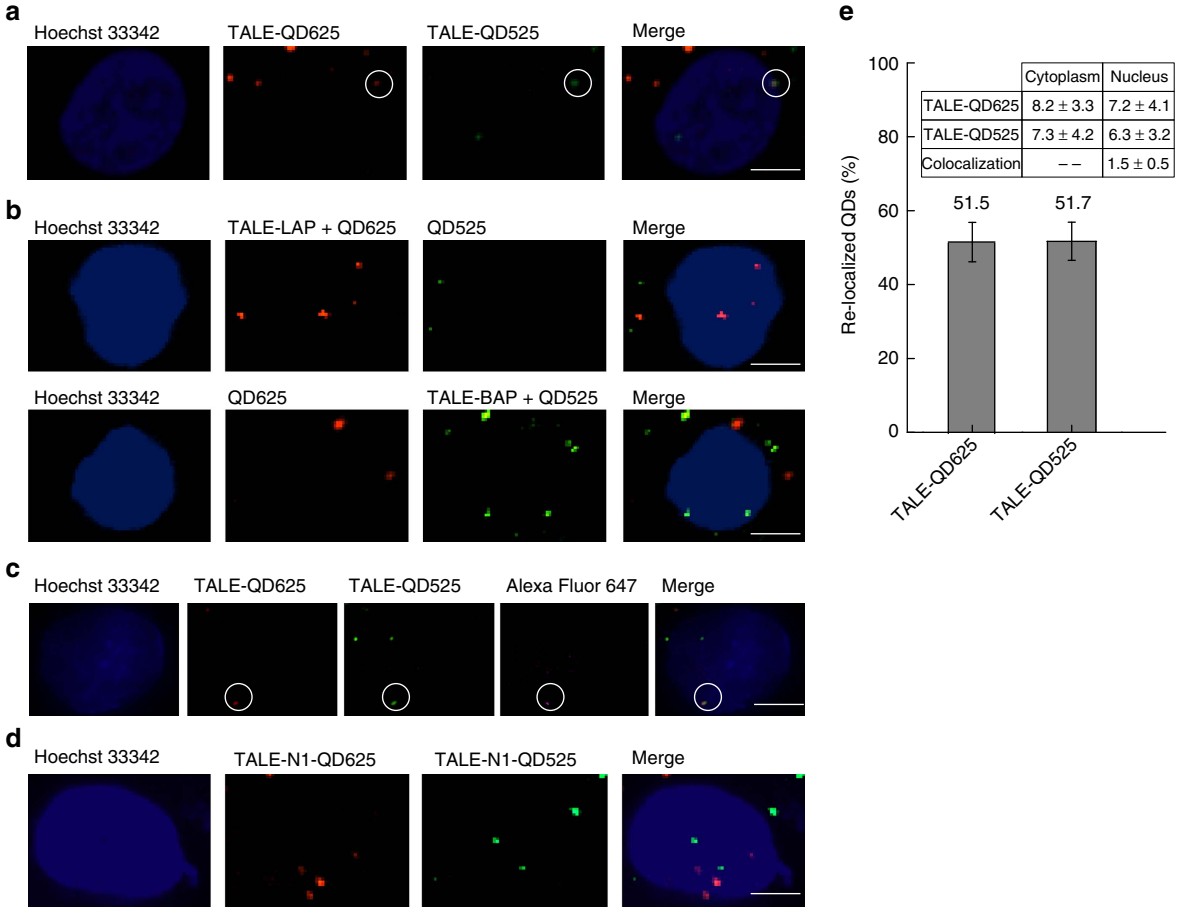

**Figure 6 | Imaging of single gene loci in live U1 cells.** (**a**) Colocalization of the N2-L-QD625 and N2-R-QD525 showing they bind to the same region of interest, observed in the cell nuclei of U1 cells. (**b**) Confocal images showing TALE-LAP, Tz1-QD625and SA-QD525 transfected into live cells (upper panel), and TALE-BAP, SA-QD525 and Tz1-QD625 transfected into live cells (lower panel). (**c**) Colocalization images of N2-L-QD625 and N2-R-QD525 with HIV-specific FISH using Alexa Fluor 647-tagged DNA probes. (**d**) Confocal images showing non-targeting N1-L-QD625, and N1-R-QD525 transfected into live cells (scale bar, 5 μm). (**e**) Percentage of TALE-QD625s and TALE-QD525s (2,000 QDs in 120 cells) re-localized to the nucleus in specific labelled cells. Inset: count and statistics of TALE-QD625s, TALE-QD525s and colocalization in the cytoplasm and nucleus. Error bars denote s.d.; $n = 120$.

While the two labels colocalized at each individual loci, the two separate HIV-1 provirus loci were not observed within the same section. The z-sectioning of the cell nucleus, or its Z-projection (Fig. 7b), clearly demonstrates there are two HIV-1 provirus loci in the cell. Three-dimensional images of XY-, XZ- and YZ-sections verified the colocalization of the N2-L-QD625 and N2-R-QD525 (Fig. 7c, Supplementary Movie 2). These results show that HIV-1 proviruses of single loci can be mapped in live cells. In the U1 cells with HIV-specific colocalization signals, the number of HIV-1 provirus loci in the nucleus was counted as $1.5 \pm 0.5$ in each cell ($n = 200$). The number of integrated HIV proviruses was also analysed by the immuno-DNA FISH method, and the average number of integrated HIV provirus loci was $1.7 \pm 0.3$ per cell. These results of number analysis (Supplementary Fig. 6) are consistent with the report that there are two copies of the HIV-1 proviral DNA in U1 cells[12].

As a control, we used the labelling method in the 293T cell line, which does not contain integrated HIV proviruses. In 293T cells, the QD625s and QD525s linked to TALEs via the bioorthogonal ligation reactions were still translocated into the cell nuclei, but they failed to show the colocalization signal that represents the HIV-1 provirus loci (Fig. 7d). The new labelling method was further used in OM10.1 cells, which contain one copy of the HIV provirus in each cell[18]. In OM10.1 cells, the QD625s and QD525s linked to TALEs could be translocated into the cell nuclei, and

one colocalization signal representing the HIV-1 provirus could be observed in 19.7% of individual cell nuclei (Fig. 7e).

## Discussion

In summary, we have reported a novel approach to allow the visualization of single genomic loci in live cells. This approach combines the sequence-specific recognition of TALEs with the single-particle sensitivity of QDs. Compared with zinc-finger proteins or engineered meganucleases, TALEs are easier to design and optimize for the purpose of sequence-specific binding and labelling. Theoretically, TALE-based techniques can be easily applied to any sequence because of the flexible design and the extreme sequence specificity of TALEs[5,19,20]. QDs have unique optical properties including remarkable brightness and excellent photostability compared with traditional organic dyes and fluorescent proteins[14]. This enables QDs to act as superior fluorophores in applications such as the ultrasensitive and long-term imaging of biological events. In this work, we have used the intense brightness of QDs to extend TALE-mediated genome visualization techniques, enabling the detection of single genomic loci. Therefore, our work provides a robust and flexible method for live cell imaging of single genomic loci.

In our study, we designed and selected a pair of TALEs for targeting adjacent sequences of the HIV-1 proviral DNA

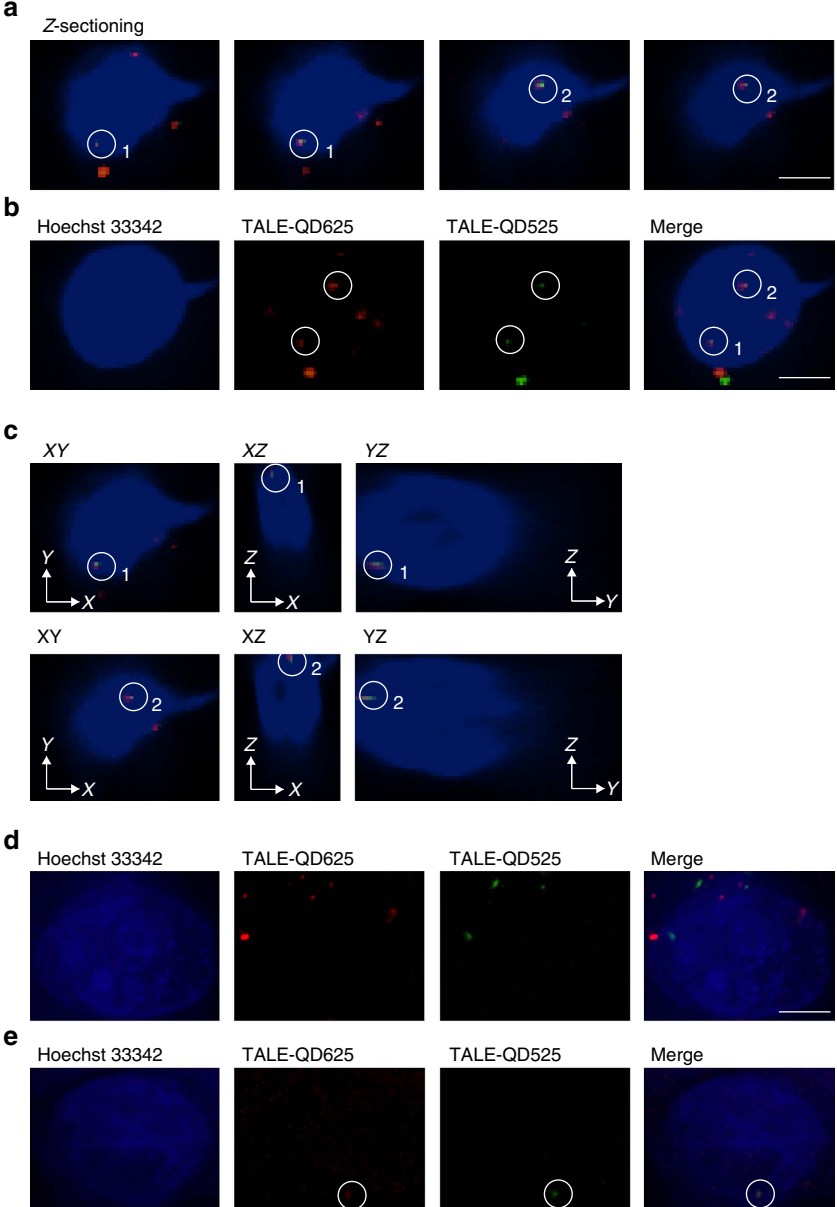

**Figure 7 | HIV-1 proviral DNAs were mapped in live U1 cells.** (**a**) Optical sections of colocalization signals were taken along the z-axis. These images were selected from 41 of z-axis slices, step size: 0.2 μm. (**b**) Colocalization signals of TALE-LAP-QD625 and TALE-BAP-QD525 were assessed in a projection image of the living cell. (**c**) The two colocalized signals of TALE-LAP-QD625 and TALE-BAP-QD525 in co-transfected U1 cells are shown in corresponding XY-, XZ- and YZ-sections. (**d**) Confocal images showing TALE-LAP-QD625 and TALE-BAP-QD525 were transfected into 293T cells. (**e**) Colocalization signal of TALE-LAP-QD625 and TALE-BAP-QD525 was assessed in OM10.1 cells. The colocalization signals are highlighted by white circles. Scale bar, 5 μm.

integrated within the chromosomes of human U1 cells. The two selected TALEs were labelled with different coloured QDs via two bioorthogonal ligation reactions. One TALE was labelled with red QD625 using an inverse electron-demand Diels–Alder cycloaddition between *trans*-cyclooctenes and tetrazines[13]. The Diels–Alder cycloaddition has been previously shown to be a fast and biocompatible reaction for labelling cellular proteins with organic dyes. Here, we show for the first time that the Diels–Alder cycloaddition can be used for labelling TALEs with QDs in live cells. A second TALE was labelled with green QD525 using an AP tag in conjunction with a biotin–streptavidin system that has been previously demonstrated in labelling proteins and viruses[14]. The TALE-conjugated QDs were successfully transported into the cell nuclei to specifically target genomic loci. Combining use of the two labelling systems permits the visualization of the two TALEs with different QDs within a single cell. The colocalization of red N2-L-QD625 and green N2-R-QD525 indicated that they were targeting the same HIV-1 proviral DNA. These results allowed single HIV-1 proviral loci to be determined and discriminated from other single-colour QD-TALEs. In our experiments, the two QD labelling systems for the pair of TALEs exhibit no cross-reactivity owing to their different reaction properties. They are biocompatible and orthogonal to each other, and thus may be widely used for the specific dual-colour labelling of any two different proteins.

Using our live cell genomic labelling technique we were able to map the HIV-1 provirus loci in live U1 cells. Two HIV-1 provirus

loci were imaged and determined within one U1 cell. This imaging result is in accordance with a previously published report, suggesting that two copies of HIV-1 proviral DNAs are integrated within the chromosomes of the U1 cell[12]. As there are only two proviral loci within one cell, these small and scarce loci are relatively difficult to specifically localize. The intense brightness of the QDs couple with three-dimensional microscopy aided us in mapping the proviral loci within a single cell. Consequently, we believe that our QD-TALEs approach offers a powerful tool for the detection and localization of single genomic loci, particularly for cases where only a few copies of single genomic loci occur in each cell. Of course, selecting a pair of TALEs with effective binding activity is a required step for successful labelling of target genomic loci. The non-targeting TALEs failed to show an HIV-specific signal in U1 cells. As our system allows live cell imaging, it offers the potential to study both the genome architecture and the dynamics of genomic loci. Thus, this novel method for examining genomic loci provides a tool for elucidating the dynamics and mechanisms of single genomic loci.

In conclusion, we have developed a method for the live cell imaging of single genomic loci by combining TALE technology with QD labelling techniques. With a pair of different coloured TALE-QDs acquired via two bioorthogonal ligation reactions, single HIV-1 proviral loci were imaged and mapped in live U1 cells. This ultra-sensitive analysis may contribute to the study of chronic HIV-1 infections, as well as virus detection and treatments at the live cell level. Our method represents a powerful and flexible approach for the direct visualization of single genomic loci in live cells, helping to elucidate their functional relevance and regulatory mechanisms in genome architecture in both healthy and disease states.

## Methods

**Construction of TALE expression plasmids.** TALEs were synthesized via 'Unit Assembly'[21,22]. The assembled repeat-variable diresidues (Beijing ComWin) were cloned into pCS2-FokI destination plasmids. For the yeast-based reporter assay, TALEs were subcloned into the eukaryotic expression vectors pTAL3 or pTAL4 (Addgene)[15]. For live imaging of target sequences, one pair of entire TALEs was amplified by PCR, and the coding sequences inserted into the mammalian expression vector pCDNA3.1(+) using BamHI and HindIII. The LAP tag or BAP tag was fused with the TALE sequences during PCR. The target sequences for all pairs of TALEs are provided in Supplementary Fig. 1.

**Yeast-based reporter assay.** By using histidine and leucine prototrophy for selection, one pair of TALEs, in pTAL3 and pTAL4, was transformed into the yeast strain YPH500 (Shanghai Genemybio)[15]. The target sequence was cloned between the lacZ fragments in pCP5 (Addgene), and transformed into yeast strain YPH499 (Shanghai Genemybio), using tryptophan and uracil prototrophy for selection. YPH500 and YPH499 were cultured in synthetic complete medium lacking histidine and leucine or tryptophan and uracil. β-Galactosidase activity was measured by mating TALE and target transformants. The cultures were adjusted to the same $A_{600}$, and 100 µl of each was added to 1 ml of yeast peptone dextrose medium and incubated for 4–6 h at 30 °C. Cells were collected by centrifugation and enzyme activity was normalized to cell number[23].

**Cell culture and transfection.** U1 (provided by Prof. Qinxue Hu, Wuhan Institute of Virology, Chinese Academy of Sciences) and OM10.1 cells (provided by Prof. Yongtang Zheng, Kunming Institute of Zoology, Chinese Academy of Sciences) were cultured in RPMI-1640 medium (Gibco) containing 10% heat inactivated FBS (Gibco). The HIV provirus DNA in U1 and OM10.1 was verified by PCR and sequencing. 293T cells (obtained from the Type Culture Collection of the Chinese Academy of Sciences, Shanghai) were maintained in DMEM (Gibco) with 10% FBS (Gibco). The cell lines have no mycoplasma contamination. Plasmids and QDs were transfected using Nucleofector (Lonza) or TransIn EL Transfection Reagent (Transgene Biotech) in accordance with the manufacturer's protocol. First, the plasmids pcDNA3-LplA ($W37V$) (1 µg, obtained from Addgene) and/or pcDNA3.1 (+)-BirA (3 µg) were transfected into $10^6$ cells cultured in a confocal dish. After incubation overnight, the plasmids pcDNA3.1 (+)-TALE-N2-L (2 µg) and/or pcDNA3.1 (+)-TALE-N2-R (2 µg), and Tz1-QD625 (100 µM) and/or SA-QD525 (150 µM) were transfected into these cells. TCO2 (200 µM) and/or biotin (50 µM) were added into the culture medium after transfection. Following additional

incubation for 6–12 h, the cells were cultured and imaged under a microscope equipped with a heated chamber at 37 °C with 5% $CO_2$. TCO2 and Tz1 were synthesized following Liu et al.[13]

**FISH.** Cells were fixed in 4% PFA at 4 °C overnight. Fixed cells were mounted onto slides at 56 °C for 1 h. Then, cells were permeabilized in 0.2% Triton X-100 for 6 min; for detection of DNA, cells were treated with RNase (Sigma-Aldrich) at a concentration of 100 µg ml$^{-1}$ in PBS for 30 min at 37 °C. After equilibration in 2 × SSC for 30 min at 37 °C, cells were dehydrated in an ethanol series (70, 85 and 100% ethanol for 2 min at room temperature), and air-dried. The plasmid pNL4-3, which contains a 9.7 kb recombinant sequence of HIV-1 DNA, was labelled with biotin-16-dUTP by nick translation and used as the DNA probe. The DNase concentration was adjusted to yield probe DNA with a fragment length of 200–500 bp[16,17]. Probe DNA was diluted to a final concentration of 10 ng µl$^{-1}$ in hybridization buffer; probe and cells were simultaneously heated at 80 °C for 10 min to denature DNA, then chilled at 0 °C for 5 min, and incubated overnight at 37 °C. After hybridization, specimens were washed with 0.3% NP-40/0.4 × SSC (for 2 min at 68 °C), 0.1% NP-40/2 × SSC (for 30 s at room temperature), 70% ethanol (for 3 min at room temperature) and air-dried. Hybridized probes were labelled with Alexa Fluor 647-strepavidin (Invitrogen), and signals were amplified with biotinylated anti-streptavidin (Vector Laboratories), followed by another round of Alexa Fluor 647-strepavidin staining. Finally, cells were stained with DAPI (1 µg ml$^{-1}$).

**Live imaging.** The cells were imaged by an UltraView VOX Confocal system (PerkinElmer, Co.) using a × 60, 1.4 NA, oil immersion objective lens. These images were acquired using an EMCCD (Hamamatsu C9100-14) with an excitation and emission filter (UltraView Dichroic), (laser power: 10 mW cm$^{-2}$, exposure: 100–200 ms). QD525s were excited with 488-nm lasers and detected with a 525-nm (W50) emission filter. QD625s were excited with 561-nm lasers and detected with a 615-nm (W70) emission filter. Alexa Fluor 647 were excited with 640-nm lasers and detected with a 705-nm (W90) emission filter. Hoechst 33342 and DAPI were excited with 405-nm lasers and detected with a 445-nm (W60) emission filter. Imaging data were analysed using Volocity 6.3.0 software. For quantitative analysing colocalization signals, regions of interest were drawn around the fluorescence dots and analysed using the Volocity software. The threshold of Pearson correlation coefficients (PCC) for the regions of interest was calculated. It has been stated that a threshold of PCC > 0.5 has a meaningful colocalization[24,25]. In our experiments, it was qualified as colocalization when PCC > 0.7.

**Statistical analysis.** Statistical analysis was performed by counting the number of QDs ($n = 2,000$) in the cytoplasm and nucleus of each cell with or without TALE conjugation systems, and the percentage of QDs that re-localized to the nucleus by TALEs was analysed. The percentages of cells showing dual-colour colocalization signals were analysed for U1 cells or OM10.1 cells ($n = 1,000$). The number of HIV-1 provirus loci in the nucleus was counted in U1 cells ($n = 200$) showing dual-colour colocalization signals. The non-targeting TALEs were also analysed in U1 cells ($n = 1,000$).

**Data availability.** The authors declare that the data supporting the findings of this study are available within the article and its Supplementary Information, or from its author upon reasonable request.

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

## Acknowledgements

Z.C. is supported by the National Natural Science Foundation of China (NSFC) (Nos 31470269 and 81590760) and the Youth Innovation Promotion Association CAS. Z.Z. is supported by NSFC (No 31470837). X.-E.Z. is grateful for support from the Chinese Academy of Sciences (KLZD-EW-TZ-L04). T.T. is supported by NSFC (Nos 21390202 and 21436002). We thank the Core Facility and Technical Support, Wuhan Institute of Virology for excellent technical support. We thank Guoqiang Wu for the cell culture and plasmids construction. We also thank the help and suggestions from Prof. Daiwen Pang (Wuhan University) and Prof. Qinxue Hu (Wuhan Institute of Virology).

## Author contributions

Z.C. and X.-E.Z. conceived and designed the experiments. Y.M. and M.W. performed the experiments. W.L. and Z.Z. analysed the data. X.Z. and T.T. contributed reagents/materials/analysis tools. Z.C. and Y.M. wrote the paper.

## Additional information

**Competing interests:** The authors declare no competing financial interests.

**Publisher's note**: 

