## [Peer Review File · Nature Communications]

Reviewers' comments:

Reviewer #1, an expert in HIV-1 integration (Remarks to the Author):

The authors report a method to visualize single loci in living cells using a combination transcription activator-like effectors (TALES) and Quantum Dots (QDs). The method is potentially useful if it can be validated, but the data presented are deficient.

1. All the data is based on imaging of a single cell. It is essential to present data in a way that allows the reader to evaluate how significant the results are. For example, data should be obtained for a population of individual cells (ideally reasonably large) for each experiment and the distribution of the data and associated errors shown.
2. It seems surprising that the background from transfecting QDs is not higher. Is it typical that only a few quantum dots enter each cell?
3. Controls should be included with cells that do not contain integrated HIV proviruses and cells that contain different known numbers of HIV proviruses.

Reviewer #2, an expert in CRISPR genome imaging (Remarks to the Author):

Cui et al combined the TALE with quantum dots to label single copy HIV DNA loci in living cells in a creative way. Non-repetitive genomic imaging are challenging due to insufficient fluorescent signal generated per DNA binding event. Previous work achieved non-repetitive genomic imaging with CRISPR/Cas9 system targeting multiple sequences tiling a genomic locus (Chen et al 2013). The authors in this study aim to achieve the same goal by enhancing the fluorescent signal generated by per Tale molecule through conjugating TALE with brighter quantum dots. Two orthogonal chemical reactions were used to conjugate a pair of TALES to quantum dots with different colors, and co-localizations of the two TALE-quantum dots are considered a true positive signal for the targeted HIV DNA locus. The idea and methods that the authors employed in this study are of great interest to people studying chromatin dynamics. However, the manuscript is suffering from major and minor issues that may prevent it from publication in the current format.

1. The authors claimed in several places that "visualization of single genomic loci" in living cells is currently "unavailable", including abstract line 2-3, the first paragraph in the main text --page 3 line 4-5 & line 10, discussion page 10 line 3. These statements are inaccurate. They have overlooked previous work that used CRISPR/Cas9 technique to visualize single non-repetitive locus in living cells (Chen et al, 2013). In this work, an array of sgRNAs targeting different non-repetitive sequences within a genomic locus is introduced into the cells to bring multiple dCas9-GFPs targeting a non-repetitive locus, thereby achieving visualization of non-repetitive locus in living cells.
2. Data validation issue. The authors use co-localizations of the two TALE-quantum dots in the nuclear to distinguish the HIV TALE targeting genomic locus from the noise (such as unbound TALE-QDs, QDs that dissociated from TALES, QD aggregations). To confirm that TALE-quantum dots co-localizing sites are the real targeting DNA locus rather than random aggregations, they need to use other established methods to validate their data. As their cells were fixed for analysis, FISH (fluorescent in situ hybridization) might be appropriate to validate that the TALE-quantum dots co-staining sites are real TALE-targeting HIV DNA loci.
3. Control cells. The authors used U1 cells that containing the TALE targeted HIV loci for

experiments. It might be good to include a control in this study using cells that lack the targeted HIV loci, in which we may expect to observe that the quantum dots can still be translocated into the nucleus when bound to NLS-TALEs, but failed to co-localize with each other (no signals). Comparing the experimental cells (U1 cells) with control cells will provide strong support that this TALE-QDS method can detect HIV loci in living cells, and more statistical data as specified below will help to evaluate the efficacy of this method.

4. Comparing figure 3A and figure 3D, it seems that there are a large abundance of TALE-GFP in the nucleus in figure 3D. Why there are only a few quantum dots targeted by TALE within the nucleus in figure 3A? Both GFP and quantum dots label TALEs, they may exhibit a similar localization pattern. How can the author explain the discrepancy? Is this a concentration dependent effect? What if they increase the concentration of quantum dots?

5. Statistic analysis issues. In figure 5 and figure 6, the authors show that nice co-localization of TALE-QD625 and TALE-QD525 dots that may represent their targeted HIV genomic loci in living cells. It is a little surprising observing so few noises per channel other than the co-localization signals, given the abundant nuclear localization of TALE-GFP shown in figure 3D. Can the author quantify the percentage of cells showing dual-color localization signals? In those cells showing positive co-localization signals, what's the statistics of noise-to-signal ratio within nucleus in each channel? Will the signal to noise ratio change when they vary the concentration of quantum dots? How many co-localizing dots are observed on average in a cell, 2 dots corresponding to the 2 copies of HIV loci? Histograms on how much dots observed per cells and better statistics will provide more information to evaluate the efficacy of the current method.

Similarly, the authors showed that quantum dots can be re-localized to the nucleus only when they are conjugated to nuclear targeting TALEs, as shown in figure 3 and 4. Can the authors better quantify this TALE-dependent quantum dots re-localization effects? For example, what percentage of cells shows the nuclear localization of quantum dots in each condition? When the quantum dots are not conjugated with TALE, is it 0 percentage? In the cells that shows nuclear re-localization, what's the average percentage of quantum dots localized in the nucleus/cytoplasm?

6. The authors aim to image single non-repetitive genomic locus in live cells, but they haven't provided a live movie in the submission. The manuscript may be improved by incorporating a time-lapse movie of TALE-QDs labeled genomic loci in living cells. The authors can use a BFP-histone construct to distinguish the nuclear signals from the cytoplasmic signals.

Minor issues:

1. How strong are the TALE-QD bindings? Is there a possibility that QD detached from TALE after being translocated into the nucleus?

2. In figures using Hoechst 33342 to stain the nucleus, the blue channels look too bright in many figures. Could the authors play with the contrast to show more details of the nuclear structure? This may help to better distinguish the nuclear localization of fluorescent signals from artifacts.

3. Can the authors explain better why QDs coupling to Tz1 induce a massive gel shift as shown in figure 2C, due to charge or molecule weight change?

4. Scale bar is missing in figure 6C. The authors should better label the XY, XZ and YZ panels. where is X, Y, Z axis, respectively? Is each row from the same cell? How thick is the Z-section? Adding 3D reconstitution will be nice.

5. Given that this is a method paper, the authors could provide more technique details of their experiments, such as the concentration of their reagents, especially the concentration of QDs. The concentration of different components may affect the results significantly.

6. Page 3, line 17-19. "integration of HIV-1 proviral DNAs into human chromosomes represents a major obstacle that has meant that diseases, such as AIDS, remain incurable." This sentence is unclear. Please revise it.

Reviewer #3, an expert in CRISPR genome imaging (Remarks to the Author):

This manuscript describes the development of an approach to label and image a genomic locus with non-repetitive sequence using quantum dot (QD) labeled TALE protein. The major innovations are: (1) the use of enzymatic tags and chemical ligation to label TALE with quantum dots, and (2) the use of two-color colocalization to determine specific labeling by TALE-QD. If the method described in the manuscript is specific, efficient and robust, it would be very useful in the study of chromosome dynamics. However, the manuscript provides no statistical analysis of these aspects, and contains too little method details for the reproduction of the results by other labs. Therefore, in current form of the manuscript, the validity of its results is questionable, and it requires major revision to be qualified for publication in Nature Communications.

The reviewer requests that the following questions be addressed by unbiased, objective statistical analysis:

1. What is the labeling specificity? A DNA FISH experiment must be performed in order to show that the colocalized QB spots are specific. How many more quantum dots (average and standard deviation) are observed in the nucleus than the actual number of target sites? Ideally, a negative control experiment using a non-targeting TALE should also be performed.

2. What is the labeling efficiency? What is the number of quantum dots in the nucleus and cytoplasm (mean and standard deviation, distribution preferred) with or without attaching to TALE? What is the fraction of cells showing specific labeling? With the DNA FISH data, one could find out all the possible binding sites. What is the fraction of these binding sites labeled by quantum dots? If these fractions are too low (e.g. < 10% of the cells have specific labeling, or < 50% of the binding sites are labeled), this QD-TALE method would not be efficient enough for practical use.

3. How robust is the method? Is activity characterization required prior to performing labeling? How would one of the less active TALEs made in the manuscript perform for labeling?

4. What is the expression level of TALE? How many copies of TALE are there in a cell and what is the fraction labeled with QD? If this fraction is low, would unlabeled TALE serve as a competitor to drive down the labeling efficiency? If the TALE expression level is as high as shown in Figure 3D, the reviewer would not believe that the observed labeling in Figure 5 is valid.

The reviewer also requests to add in the following experimental conditions: Concentration of QD for nucleofection, concentration of TCO2 and biotin used for labeling, filters and objective (model, numerical aperture, etc.) used for microscopy, camera model and exposure parameters for image acquisition, and other essential experimental details.

Reviewers' comments:

Reviewer #1, an expert in HIV-1 integration:

The authors report a method to visualize single loci in living cells using a combination transcription activator-like effectors (TALEs) and Quantum Dots (QDs). The method is potentially useful if it can be validated, but the data presented are deficient.

Response: Thank you for the comments and suggestions. We have revised our manuscript according to your suggestions.

1. All the data is based on imaging of a single cell. It is essential to present data in a way that allows the reader to evaluate how significant the results are. For example, data should be obtained for a population of individual cells (ideally reasonably large) for each experiment and the distribution of the data and associated errors shown.

Response: Yes, we agree and have included images with more cells instead of only a single cell (Please see Fig. 3A-3B, Fig. 4A-4B, and Supplementary Fig. 3). We have also added statistical analyses of the data that was obtained for a population of individual cells in our revised manuscript (Please see Fig. 3E, Fig. 4C, and Fig. 5E; Page 6, Lines 9-11, Page 7, Lines 8-9, Page 8, Lines 9-12, and Page 9, Lines 3-4).

2. It seems surprising that the background from transfecting QDs is not higher. Is it typical that only a few quantum dots enter each cell?

Response: There is a background from transfecting QDs. In our experiment, to achieve a relatively high labeling efficiency and a low background level during imaging, we applied an optimal experimental condition with two transfection steps: cells were first transfected with Lp1A and BirA, and cultured overnight, and then transfected with QDs (100 μ M of Tz1-QD625 and 150 μ M of SA-QD525) and TALEs. If higher concentration of QDs were used during transfection, more quantum dots would be transfected into the cell, but the background would also increase accordingly.

3. Controls should be included with cells that do not contain integrated HIV proviruses and cells that contain different known numbers of HIV proviruses.

Response: We agree and have included these controls in our revised manuscript (Fig. 6D-6E, Page 9, Lines 5-12).

1) We have included a control with the 293T cells that do not contain integrated HIV proviruses. In 293T cells, the red and green quantum dots linked to TALEs via bioorthogonal ligation reactions could be translocated into the cell nuclei, but failed to show the colocalization signal representing the HIV-1 proviral DNA (Please see Fig. 6D, Page 9, Lines 5-8).

2) We have also included the imaging results from OM10.1 cells that contain one copy of HIV provirus in each cell. In OM10.1 cells, the red and green quantum dots linked to TALEs could be translocated into the cell nuclei, and one colocalization signal representing the HIV-1 proviral DNA can be observed in the cell nucleus (Please see Fig. 6E, Page 9, Lines 8-12, ref. 16).

Reviewer #2, an expert in CRISPR genome imaging:

Cui et al combined the TALE with quantum dots to label single copy HIV DNA loci in living cells in a creative way. Non-repetitive genomic imaging are challenging due to insufficient fluorescent signal generated per DNA binding event. Previous work achieved non-repetitive genomic imaging with CRISPR/Cas9 system targeting multiple sequences tiling a genomic locus (Chen et al 2013). The authors in this study aim to achieve the same goal by enhancing the fluorescent signal generated by per Tale molecule through conjugating TALE with brighter quantum dots. Two orthogonal chemical reactions were used to conjugate a pair of TALEs to quantum dots with different colors, and co-localizations of the two TALE-quantum dots are considered a true positive signal for the targeted HIV DNA locus. The idea and methods that the authors employed in this study are of great interest to people studying chromatin dynamics. However, the manuscript is suffering from major and minor issues that may prevent it from publication in the current format.

Response: Thank you for the comments and suggestions. We have revised our manuscript according to your suggestions. Detailed point-by-point responses to your comments are given below.

1. The authors claimed in several places that "visualization of single genomic loci" in living cells is currently "unavailable", including abstract line 2-3, the first paragraph in the main text --page 3 line 4-5 & line 10, discussion page 10 line 3. These statements are inaccurate. They have overlooked previous work that used CRISPR/Cas9 technique to visualize single non-repetitive locus in living cells (Chen et al, 2013). In this work, an array of sgRNAs targeting different non-repetitive sequences within a genomic locus is introduced into the cells to bring multiple dCas9-GFPs targeting a non-repetitive locus, thereby achieving visualization of non-repetitive locus in living cells.

Response: We agree and we have already changed these statements. For example, we have changed "unavailable" to "challenging" in the revised manuscript. Please also see our changes in the main text -- page 2 line 3, page 3 line 5 and lines 10-11, as well as in the discussion on page 10 line 3. We appreciated the Chen et al's work very much, in which they have used multiple GFP-labeled dCas9 probes to visualize of non-repetitive locus in living cells. Here, we have combined the optical superiority of QDs with the sequence-specific recognition of TALEs to realize the visualization of single non-repetitive genomic loci in live cells.

2. Data validation issue. The authors use co-localizations of the two TALE-quantum dots in the nuclear to distinguish the HIV TALE targeting genomic locus from the noise (such as unbound TALE-QDs, QDs that dissociated from TALEs, QD aggregations). To confirm that TALE-quantum dots co-localizing sites are the real targeting DNA locus rather than random aggregations, they need to use other established methods to validate their data. As their cells were fixed for analysis, FISH (fluorescent in situ hybridization) might be appropriate to validate that the TALE-quantum dots co-staining sites are real TALE-targeting HIV DNA loci.

Response: We agree and have included FISH (fluorescent in situ hybridization) to validate that the TALE-quantum dots co-staining sites are real TALE-targeted HIV DNA loci rather than random aggregations. As shown in Fig. 5C and Supplementary Fig. 5, we observed the co-localization of TALE-QDs with the HIV DNA FISH sites labeled by QD705-tagged HIV-specific DNA probes (Page 8, Lines 2-6).

3. Control cells. The authors used U1 cells that containing the TALE targeted HIV loci for experiments. It might be good to include a control in this study using cells that lack the targeted HIV loci, in which we may expect to observe that the quantum dots can still be translocated into the nuclear when bound to NLS-TALEs, but failed to co-localize with each other (no signals). Comparing the experimental cells (U1 cells) with control cells will provide strong support that this TALE-QDs method can detect HIV loci in living cells, and more statistical data as specified below will help to evaluate the efficacy of this method.

Response: We agree and have included two additional control cells in our revised manuscripts. One control cell is the 293T cell that does not contain integrated HIV proviruses. In 293T cells, the red and green quantum dots linked to TALEs via bioorthogonal ligation reactions translocated into the cell nuclei, but failed to show a colocalization signal representing the HIV-1 proviral DNA (Please see Fig. 6D, Page 9, Lines 5-8).

The other control is the OM10.1 cell line that contains one copy of HIV provirus in each cell. In OM10.1 cells, the red and green quantum dots linked to TALEs translocated into the cell nuclei, and one colocalization signal representing the HIV-1 proviral DNA can be observed in the cell nucleus (Please see Fig. 6E, Page 9, Lines 8-12, ref. 16).

4. Comparing figure 3A and figure 3D, it seems that there are a large abundance of TALE-GFP in the nucleus in figure 3D. Why there are only a few quantum dots targeted by TALE within the nucleus in figure 3A? Both GFP and quantum dots label TALEs; they may exhibit a similar localization pattern. How can the author explain the discrepancy? Is this a concentration dependent effect? What if they increase the concentration of quantum dots?

Response: TALE-GFP was expressed in the cell by transfection of a plasmid (pEGFP-TALE) into the cell. The amount of TALE-GFP protein increased during the protein expression. This experiment was performed to show the nuclear localization of the TALE protein because the TALE-GFP fusion protein is localized to the cell nucleus.

As for quantum dots, a given amount of quantum dots was transfected into the cells and they could not replicate and increase within the cell. Only the ODs linked to TALEs via bioorthogonal ligation reactions could be translocated into the cell nuclei by TALEs. Therefore, there may be a discrepancy between the GFP and quantum dot signals.

If we increase the concentration of QDs during transfection, more quantum dots would be transfected into the cell nucleus, but the background would also increase when labeling targets.

5. Statistic analysis issues. In figure 5 and figure 6, the authors show that nice co-localization of TALE-QD625 and TALE-QD525 dots that may represent their targeted HIV genomic loci in living cells. It is a little surprising observing so few noises per channel other than the co-localization signals, given the abundant nuclear localization of TALE-GFP shown in figure 3D. Can the author quantify the percentage of cells showing dual-color localization signals? In those cells showing positive co-localization signals, what's the statistics of noise-to-signal ratio within nucleus in each channel? Will the signal to noise ratio change when they vary the concentration of quantum dots? How many co-localizing dots are observed on average in a cell, 2 dots corresponding to the 2 copies of HIV loci? Histograms on how much dots observed per cells and better statistics will provide more information to evaluate the efficacy of the current method.

Response: We agree and have included statistical analyses in our revised manuscript. In our experiments, we acquired the result that 21.3% of cells showed dual-color localization signals. In

those cells showing positive co-localization signals, the noise-to-signal ratios within nucleus was $(4.8 \pm 2.7) : 1$ in the QD625 channel, and $(4.2 \pm 2.1) : 1$ of in the QD525 channel. The signal to noise ratio would change when we varied the concentration of quantum dots. As the concentration of QDs increased, there were more QDs transfected into the nucleus, but the noise signal would also increase.

Under the condition in our experiments, the average number of co-localized dots observed in a cell was 1.5 ± 0.5 ($n=200$). In U1 cells, we did not see more than 2 dots, corresponding to the 2 copies of HIV loci.

According to your suggestion, we have also provided histograms of the statistics. Please see the Fig. 5E, Page 8 Lines 9-12, and Page 9 Lines 3-4.

Similarly, the authors showed that quantum dots can be re-localized to the nucleus only when they are conjugated to nuclear targeting TALEs, as shown in figure 3 and 4. Can the authors better quantify this TALE-dependent quantum dots re-localization effects? For example, what percentage of cells shows the nuclear localization of quantum dots in each condition? When the quantum dots are not conjugated with TALE, is it 0 percentage? In the cells that shows nuclear re-localization, what's the average percentage of quantum dots localized in the nucleus/cytoplasm?

Response: We agree and have quantified the TALE-dependent quantum dot re-localization effects. In our experiments, 54.5% of cells showed the nuclear signals of QD625s (number: 9.2 ± 4.1 , Fig. 3E) being re-localized via the Diels–Alder cycloaddition system, and 57.1% of cells showed the nuclear signals of QD525s (number: 8.4 ± 4.2 , Fig. 4C) being re-localized via the biotin-streptavidin system, and 21.3% of transfected cells showed the colocalized signals.

When quantum dots were conjugated with TALE, the percentage of cells with the nuclear quantum dot signals was 0 (Fig. 3B and Fig. 4B and Fig. 5B and Supplementary Fig. 4). In the cells that showed nuclear re-localization, the average percentages localized in the nucleus/cytoplasm were 56.4% and 43.6%, respectively, for QD625s re-localized via the Diels–Alder cycloaddition system and 53.2% and 46.8%, respectively, for QD525s re-localized via the biotin-streptavidin system (Fig. 3E and Fig. 4C, Page 6, Lines 9-11, and Page 7, Lines 8-9).

6. The authors aim to image single non-repetitive genomic locus in live cells, but they haven't provided a live movie in the submission. The manuscript may be improved by incorporating a time-lapse movie of TALE-QDs labeled genomic loci in living cells. The authors can use a BFP-histone construct to distinguish the nuclear signals from the cytoplasmic signals.

Response: Yes, we acquired some time-lapse images of the genomic locus in our experiments. However, we have not included these results in the manuscript because we did not obtain any meaningful dynamic information. Nevertheless, the method does have the capacity to track the genomic locus in real time.

In the current study, we used Hoechst 33342 to stain the nucleus, combined with three-dimensional imaging to distinguish the nuclear signals from the cytoplasmic signals. Of course, a BFP-histone is also a good candidate to reach this goal.

Minor issues:

1. How strong are the TALE-QD bindings? Is there a possibility that QD detached from TALE after being translocated into the nucleus?

Response: In our system, QD625 was linked to TALE-LAP through a covalent bond of TCO2 (COOH) and Tz1 (NH₂), and QD525 was linked to TALE-BAP through interaction of biotin and streptavidin. These two types of connections have been shown to be strong and specific by many studies (ref. 13-14).

2. In figures using Hoechst 33342 to stain the nucleus, the blue channels look too bright in many figures. Could the authors play with the contrast to show more details of the nuclear structure? This may help to better distinguish the nuclear localization of fluorescent signals from artifacts.

Response: We agree and have played with the contrast to better distinguish the nuclear localization of fluorescent signals from artifacts.

3. Can the authors explain better why QDs coupling to Tz1 induce a massive gel shift as shown in figure 2C, due to charge or molecule weight change?

Response: In agarose gel electrophoresis, macromolecules with charge can migrate to the opposite electrode. QDs modified with carboxyl group are negatively charged, and have a massive gel shift. When QDs were coupled with Tz1, Tz1 should neutralize the negative charge of QDs, and thus, the QDs-Tz1 complex has almost no gel shift.

4. Scale bar is missing in figure 6C. The authors should better label the XY, XZ and YZ panels. Where is X, Y, Z axis, respectively? Is each row from the same cell? How thick is the Z-section? Adding 3D reconstitution will be nice.

Response: We agree and have added the scale bar and labeled the XY, XZ and YZ panels in Fig. 6. Fig. 6A-6C is from the same cell. Images in Fig. 6A were selected from 41 of z-axis slices, step size: 0.2 μm . We also added the 3D reconstitution (supplementary Movie 1).

5. Given that this is a method paper, the authors could provide more technique details of their experiments, such as the concentration of their reagents, especially the concentration of QDs. The concentration of different components may affect the results significantly.

Response: We agree and have included more experimental details in our manuscript, including the concentration of QDs for nucleofection, the concentration of TCO2 and biotin used for labeling, filters and objective (model, numerical aperture, etc.) used for microscopy, camera model and exposure parameters for image acquisition (Page 12 Lines 14-26, and Page 13 Lines 1-25).

6. Page 3, line 17-19. "Integration of HIV-1 proviral DNAs into human chromosomes represents a major obstacle that has meant that diseases, such as AIDS, remain incurable." This sentence is unclear. Please revise it.

Response: We have revised this sentence to "It is recognized that the integration of HIV-1 proviral DNA into human chromosomes represents a major obstacle to eradicating the virus, which makes AIDS a difficult disease to cure".

Reviewer #3, an expert in CRISPR genome imaging:

This manuscript describes the development of an approach to label and image a genomic locus with non-repetitive sequence using quantum dot (QD) labeled TALE protein. The major innovations are: (1) the use of enzymatic tags and chemical ligation to label TALE with quantum dots, and (2) the use of two-color colocalization to determine specific labeling by TALE-QD. If the method described in the manuscript is specific, efficient and robust, it would be very useful in the study of chromosome dynamics. However, the manuscript provides no statistical analysis of these aspects, and contains too little method details for the reproduction of the results by other labs. Therefore, in current form of the manuscript, the validity of its results is questionable, and it requires major revision to be qualified for publication in Nature Communications.

Response: Thank you for the comments and suggestions. We have revised our manuscript according to your suggestions.

1. What is the labeling specificity? A DNA FISH experiment must be performed in order to show that the colocalized QD spots are specific. How many more quantum dots (average and standard deviation) are observed in the nucleus than the actual number of target sites? Ideally, a negative control experiment using a non-targeting TALE should also be performed.

Response: 1) We agree and have included FISH (fluorescent in situ hybridization) to validate that the TALE-quantum dots co-staining sites are real TALE-targeted HIV DNA loci rather than nonspecific aggregations. As shown in Fig. 5C and supplementary Fig. 5, we observed the colocalization of the colocalized TALE-QDs with HIV-specific FISH sites stained by QD705-tagged DNA probes (Page 8, Lines 2-6).

2) We have also included statistical analyses of target signals and free QDs in the nucleus. Under our current experiment condition, the number of QD625, QD525, and colocalization signals in the nucleus were 7.2 ± 4.1 , 6.3 ± 3.2 , and 1.5 ± 0.5 ($n=200$) per cell, respectively (Fig. 5E).

3) We have included a negative control experiment using a pair of non-targeting TALEs. As shown in Fig. 5D, the QD625s and QD525s could still be translocated into the nucleus by linking to the non-targeting TALEs, but they failed to co-localize with each other to show an HIV specific signal (Page 8, Lines 6-9).

2. What is the labeling efficiency? What is the number of quantum dots in the nucleus and cytoplasm (mean and standard deviation, distribution preferred) with or without attaching to TALE? What is the fraction of cells showing specific labeling? With the DNA FISH data, one could find out all the possible binding sites. What is the fraction of these binding sites labeled by quantum dots? If these fractions are too low (e.g. < 10% of the cells have specific labeling, or < 50% of the binding sites are labeled), this QD-TALE method would not be efficient enough for practical use.

Response: Thanks for your suggestion. We have included statistical analyses of the efficiency in our revised manuscript.

In our experiments, the number of QD625s in the nucleus and cytoplasm with attached to TALE are 9.2 ± 4.1 and 7.1 ± 3.2 per cell, which means that 56.4% of QD625s were re-localized to the nucleus by TALE and the Diels–Alder cycloaddition system. The number of QD525s in the nucleus and cytoplasm with attached to TALE are 8.4 ± 4.2 and 7.4 ± 3.3 per cell, which means that 53.2% of QD525s were re-localized to the nucleus by TALE and the biotin-streptavidin system (Fig. 3E and Fig. 4C). When quantum dots were not conjugated with TALE, the

percentage of quantum dots localized to the nucleus was 0 (Fig. 3B and Fig. 4B and Fig. 5B and Supplementary Fig. 4).

In our experiments, 21.3% of the cells showed the HIV specific colocalization signals. And in these U1 cells, the number of colocalization signals in the nucleus was 1.5 ± 0.5 (n=200) per cell. Of course, if we increase the concentration of transfected QDs, more quantum dots could be translocated into the cell nucleus and a higher percentage of the cells would be labeled, but the noise signals would also increase (Page 6, Lines 9-11, Page 7, Lines 8-9, Page 8, Lines 9-12, and Page 9, Lines 3-4).

3. How robust is the method? Is activity characterization required prior to performing labeling? How would one of the less active TALEs made in the manuscript perform for labeling?

Response: We have selected and verified the activity of the TALE. To perform the labeling, we just used the selected TALEs with good properties. The other elements of the labeling system are constant, and thus the method should be robust.

As for the less active TALEs, we have included a negative control experiment using a pair of non-targeting TALEs. As shown in Fig. 5D, the QD625s and QD525s could still be translocated into the nucleus when linked to the non-targeting TALEs, but they failed to co-localize with each other to show an HIV specific signal (Page 8, Lines 6-9).

4. What is the expression level of TALE? How many copies of TALE are there in a cell and what is the fraction labeled with QD? If this fraction is low, would unlabeled TALE serve as a competitor to drive down the labeling efficiency? If the TALE expression level is as high as shown in Figure 3D, the reviewer would not believe that the observed labeling in Figure 5 is valid.

Response: Yes, the TALE expression level during our experiments is an important point. The copies of TALE would increase with the expression time during the transfection process. In our experiments, we have used two transfection steps to acquire a relatively low level of TALE and an optimal labeling efficiency. First, the plasmids expressing lipoic acid ligase (LplA) and biotin ligase (BirA) were transfected the cell, and after incubation overnight, the QDs and the plasmids expressing TALEs were transfected into the cell. The cells were then imaged using microscopy following further 6 h incubation. This procedure ensures that there are excessive ligases, and the early expressed TALE and the transfected QDs were linked efficiently. Through this optimization, we have achieved 21.3% efficiency of the cells showing the HIV specific colocalization signals (Page 6 Lines 5-9, and Page 8 Lines 9-12).

The reviewer also requests to add in the following experimental conditions: Concentration of QD for nucleofection, concentration of TCO2 and biotin used for labeling, filters and objective (model, numerical aperture, etc.) used for microscopy, camera model and exposure parameters for image acquisition, and other essential experimental details.

Response: We agree and have added these experimental details including the concentration of QDs for nucleofection, the concentration of TCO2 and biotin used for labeling, filters and objective (model, numerical aperture, etc.) used for microscopy, camera model and exposure parameters for image acquisition, and other essential experimental conditions (Page 12 Lines 14-26, and Page 13 Lines 1-25).

Reviewers' comments:

Reviewer #1 (Remarks to the Author):

I am satisfied with the authors revisions and responses. One minor issue: in Figures 3E, 4C, and 5E please indicate how many cells were counted.

Reviewer #2 (Remarks to the Author):

The authors have made great efforts revising the manuscripts with the added statistical analysis, essential controls and imaging data. Now the results presented in the manuscript are greatly improved and more convincing. But there are a few points the authors might want to better address before its publication.

1. The FISH result is essential to validate that the TALE-quantum dots co-staining sites are true TALE-targeting HIV DNA loci rather than random aggregations, and it's nice that the authors have added this validation. But the authors used quantum dots again for the FISH experiment, raising a concern that QDs may intrinsically tend to aggregate together in the same cellular region rather than recognizing true signals. The figure 5C shows only one dot in all three QD channels, so it is difficult to tell if the FISH probe may also localize to non-targeting QD aggregates due to intrinsic aggregation properties of QDs. Can the authors try to find ways to prove that their FISH results distinguish well between real signal and aggregations of QDs? Or is it possible to use other dyes to avoid confusion? Also the authors haven't provided enough information for the HIV FISH probe. Is it commercially available (comingTech or corningTech) or cited anywhere else (references)? Or if the authors have designed them, could they provide more detailed info (sequence, etc.)?

2. Details of statistical analysis. It would be great if the authors could provide the numbers of QDs and the numbers of cells used for each statistical analysis. Is there any statistics on the data from OM10.1 cells to compare with U1 cells?

3. Even though the authors "haven't obtain any meaningful dynamic information" from time-lapse imaging, it might still be good to include an example movie/time-lapse image as the paper is developing tools to image dynamics of genomic loci in living cells.

Reviewer #3 (Remarks to the Author):

The reviewer is glad to see the inclusion of more essential control experiments and statistical analyses in the revised manuscript, which undoubtedly strengthened the claims. Nevertheless, there still lack direct, statistical assessments of the labeling efficiency and several other important aspects of the method performance, which were originally requested by the reviewer. Without these numbers, again, the reviewer is unable to fully assess the validity and the potential impact of the reported method.

Specifically, the reviewer requests the following information:

1. Labeling efficiency: what is the fraction of integrated HIV DNA that can be detected by this method. The efficiency can be calculated through any of the following routes (preferably all of them so that there is cross-validation): (1) In U1 cells, what is the average number of integrated

HIV proviruses per cell, measured by any other independent method? By comparing this number with the 1.5 ± 0.5 colocalized QD dots per cell (the reviewer prefer reporting a histogram in addition to just the mean value), one can estimate the labeling efficiency. (2) In U1 cells, what is the fraction of FISH spots colocalized also showing three-color colocalization with the two QD channels. Assuming a 100% FISH efficiency, this number can also give an estimation of labeling efficiency. (3) In OM10.1 cells where there is only one copy of HIV provirus integration, what is the histogram of the number of colocalized QD pairs per cell? This plot will be able to tell both the labeling efficiency and specificity (when having more than one colocalized QD pair per cell). Moreover, please indicate whether the confidence bounds are standard deviation or standard error of mean.

2. Labeling specificity: The only piece of result showing specific label is that QDs on two non-targeting TALEs "fail to colocalize". How many cells have been analyzed? Has there been objective criteria used for measuring colocalization? It is evident from the images that the red and green spots are not exactly overlapping, presumably due to chromatic aberrations of the microscope. In this case, one would expect a higher probability of observing random colocalization simply because two QDs are close to each other by chance. If the colocalization is purely determined by eyes, the result could be easily biased unless a double-blind analysis was performed (i.e. the person who analyzes the colocalization should be a different person as the one who took the data, and should not know which sample the picture is from). For objective analysis, an easy way is to identify all particles in both the red and green channels, and define colocalization as red and green spots closer than a distance threshold. This distance threshold will also allow the estimation of the probability for random colocalization.

3. Labeling robustness: The reviewer was asking whether the TALE selection is a required step for successful labeling. Answering this question needs data with the less active TALEs but not the non-targeting TALEs. If such data is unavailable, at least this point needs to be mentioned in the discussion.

Minor points:

Please be more specific regarding the filter information of the microscope, which seems to be available here: http://www.zmbh.uni-heidelberg.de/Central_Services/Imaging_Facility/info/UltraVIEW.pdf

Reviewers' comments:

Reviewer #1:

I am satisfied with the author's revisions and responses. One minor issue: in Figures 3E, 4C, and 5E please indicate how many cells were counted.

Response: Thank you for your positive comment and suggestion. Cell numbers in the statistical analysis in Figures 3E, 4C, and 5E have been included in the revised manuscript (please see Page 17, Line 4; Page 17, Line 10; Page 17, Line 21).

Reviewer #2:

The authors have made great efforts revising the manuscripts with the added statistical analysis, essential controls and imaging data. Now the results presented in the manuscript are greatly improved and more convincing. But there are a few points the authors might want to better address before its publication.

Response: Thank you for the positive comments and suggestions. We have revised our manuscript according to your suggestions.

1. The FISH result is essential to validate that the TALE-quantum dots co-staining sites are true TALE-targeting HIV DNA loci rather than random aggregations, and it's nice that the authors have added this validation. But the authors used quantum dots again for the FISH experiment, raising a concern that QDs may intrinsically tend to aggregate together in the same cellular region rather than recognizing true signals. The figure 5C shows only one dot in all three QD channels, so it is difficult to tell if the FISH probe may also localize to non-targeting QD aggregates due to intrinsic aggregation properties of QDs. Can the authors try to find ways to prove that their FISH results distinguish well between real signal and aggregations of QDs? Or is it possible to use other dyes to avoid confusion? Also the authors haven't provided enough information for the HIV FISH probe. Is it commercially available (comingTech or corningTech) or cited anywhere else (references)? Or if the authors have designed them, could they provide more detailed info (sequence, etc.)?

Response: We agree and have carried out the FISH experiment by using Alexa Fluor® 647-tagged HIV-specific DNA probes to avoid the confusion. An ultra-sensitive immuno-DNA FISH experiment (Bell et al, J Virol 2001, 75:7683; Marini et al, Nature 2015, 521:227), has been performed to validate that the TALE-QDs targeted real HIV DNA loci rather than random aggregations. The HIV-1 NL4-3 DNA fragments labeled with biotin-16-dUTP were used as the probe. Hybridized probes were stained with Alexa Fluor® 647-streptavidin, and signals were amplified with biotinylated anti-streptavidin, followed by another round of Alexa Fluor® 647-streptavidin staining. As shown in Fig. 5C and Supplementary Fig. 5, co-localization of TALE-QDs with the HIV DNA FISH sites stained by Alexa Fluor® 647-tagged HIV-specific DNA probes were observed (Page 8, Lines 3 and 6-7; Page 13, Lines 14-16 and 20-22).

The information for the HIV FISH probe has also been included in the revised manuscript (Page 13, Lines 14-16 and 20-22). The plasmid pNL4-3, which contains a

9.7 kb recombinant sequence of HIV-1 DNA, was labeled with biotin-16-dUTP by nick translation and used as the DNA probe. The DNase concentration was adjusted to yield probe DNA with a fragment length of 200 to 500 bp. Hybridized probes were labeled with Alexa Fluor® 647-streptavidin (Invitrogen), and signals were amplified with biotinylated anti-streptavidin (Vector Laboratories), followed by another round of Alexa Fluor® 647-streptavidin staining.

2. Details of statistical analysis. It would be great if the authors could provide the numbers of QDs and the numbers of cells used for each statistical analysis. Is there any statistics on the data from OM10.1 cells to compare with U1 cells?

Response: Yes, we have included the number of QDs and the number of cells used for each statistical analysis in the revised manuscript (Page 17, Line 4; Page 17, Line 10; Page 17, Line 21). We have also added statistical analysis of OM10.1 cells in the revised manuscript, and the result showed that 19.7% of the cells exhibited the HIV specific colocalization signals (Page 9, Line 17).

3. Even though the authors “haven’t obtain any meaningful dynamic information” from time-lapse imaging, it might still be good to include an example movie/time-lapse image as the paper is developing tools to image dynamics of genomic loci in living cells.

Response: Thank you for the suggestion. We have included a live movie in the revised manuscript (Page 7, Lines 21-22; Supplementary Movie 1).

Reviewer #3:

The reviewer is glad to see the inclusion of more essential control experiments and statistical analyses in the revised manuscript, which undoubtedly strengthened the claims. Nevertheless, there still lack direct, statistical assessments of the labeling efficiency and several other important aspects of the method performance, which were originally requested by the reviewer. Without these numbers, again, the reviewer is unable to fully assess the validity and the potential impact of the reported method.

Response: Thank you for the comments and suggestions. We have revised our manuscript according to your suggestions.

Specifically, the reviewer requests the following information:

1. Labeling efficiency: what is the fraction of integrated HIV DNA that can be detected by this method. The efficiency can be calculated through any of the following routes (preferably all of them so that there is cross-validation): (1) In U1 cells, what is the average number of integrated HIV proviruses per cell, measured by any other independent method? By comparing this number with the 1.5 ± 0.5 colocalized QD dots per cell (the reviewer prefer reporting a histogram in addition to just the mean value), one can estimate the labeling efficiency. (2) In U1 cells, what is the fraction of FISH spots colocalized also showing three-color colocalization with the two QD channels. Assuming 100% FISH efficiency, this number can also give an estimation of labeling efficiency. (3) In OM10.1 cells where there is only one copy of

HIV provirus integration, what is the histogram of the number of colocalized QD pairs per cell? This plot will be able to tell both the labeling efficiency and specificity (when having more than one colocalized QD pair per cell). Moreover, please indicate whether the confidence bounds are standard deviation or standard error of mean.

Response: Thank you for the suggestion. According to your suggestion, we have measured the average number of integrated HIV proviruses per cell by the other independent method, an immuno-DNA FISH method (Page 8, Lines 3 and 6-7; Page 13, Lines 14-16 and 20-22) (Bell et al, J Virol 2001, 75:7683; Marini et al, Nature 2015, 521:227). The average number of integrated HIV provirus loci was 1.7 ± 0.3 in each cell of FISH experiments (Page 9, Lines 6-9), which is comparable to the 1.5 ± 0.5 colocalized QD dots per cell in the TALE-labeled method. The results from both methods are consistent with the previous report that there are 2 copies of the HIV-1 proviral DNA in U1 cells (Hu et al, PNAS 2014, 111:11461).

According to your suggestion, we have also included a histogram (Supplementary Fig. 6) to demonstrate the numbers of HIV specific signals from TALE-QDs and FISH in U1 cells, which should better display the labeling efficiency.

We have also added the analysis results of OM10.1 cells in the revised manuscript. In OM10.1 cells, one colocalization signal representing the HIV-1 provirus could be observed in 19.7% of individual cell nuclei. (Page 9, Line 17).

2. Labeling specificity: The only piece of result showing specific label is that QDs on two non-targeting TALEs “fail to colocalize”. How many cells have been analyzed? Has there been objective criteria used for measuring colocalization? It is evident from the images that the red and green spots are not exactly overlapping, presumably due to chromatic aberrations of the microscope. In this case, one would expect a higher probability of observing random colocalization simply because two QDs are close to each other by chance. If the colocalization is purely determined by eyes, the result could be easily biased unless a double-blind analysis was performed (i.e. the person who analyzes the colocalization should be a different person as the one who took the data, and should not know which sample the picture is from). For objective analysis, an easy way is to identify all particles in both the red and green channels, and define colocalization as red and green spots closer than a distance threshold. This distance threshold will also allow the estimation of the probability for random colocalization.

Response: We agree with your suggestions. We have included the cell number for analysis. The non-targeting TALEs were also analyzed in U1 cells (n=1000) (Page 14, Lines 12-13).

The fluorescence colocalization in our experiment has also been quantitatively analyzed using Volocity software. We have added the details of this analysis in the revised manuscript (Page 14, Lines 3-6). For quantitative analyzing colocalization signals, regions of interest (ROIs) were drawn around the fluorescence dots and analyzed using the Volocity software. The threshold of Pearson correlation coefficients (PCC) for the entire ROI was calculated. It has been stated that a threshold of $PCC > 0.5$ indicates a meaningful colocalization (Adler et al, Cytometry Part A 2010, 77:733; Rana et al, J Cell Biol 2015, 209:653). In our experiments, when

PCC >0.7, it was qualified as colocalization. All of the colocalization data in our manuscript meet this criterion.

3. Labeling robustness: The reviewer was asking whether the TALE selection is a required step for successful labeling. Answering this question needs data with the less active TALEs but not the non-targeting TALEs. If such data is unavailable, at least this point needs to be mentioned in the discussion.

Response: Thank you for the suggestion. We have added a discussion about this point: selecting a pair of TALEs with effective binding activity is a required step for successful labeling of target genomic loci. The non-targeting TALEs failed to show an HIV specific signal in U1 cells (Page 11, Lines 13-16).

Minor points:

Please be more specific regarding the filter information of the microscope, which seems to be available here:

http://www.zmbh.uni-heidelberg.de/Central_Services/Imaging_Facility/info/UltraVIEW.pdf

Response: The filter information of the microscope has been included in the revised manuscript (Page 13, Lines 26-28; Page 14, Lines 1-2).

REVIEWERS' COMMENTS:

Reviewer #2 (Remarks to the Author):

The authors have included more details of quantification that improved the quality of the paper. However, there are some problems in the revision that the authors need to further address.

1). There is a problem in the design of the FISH assay to validate that the TALE-QD co-labeling loci are true targeting site rather than aggregations. The authors used biotin-streptavidin interaction (biotin-HIV DNA probe and Alexa Fluor® 647-streptavidin interaction) for the FISH experiment for validating the TALE-QD signals, but this design is problematic given that the same biotin-streptavidin reaction is already been used to conjugate to TALE-biotin to streptavidin-QDs (TALE N2-R and QD525). Therefore, the FISH result is not valid due to crosstalk between the TALE-N2-R and Alexa Fluor® 647-strepavidin. The authors need to find an orthogonal way to validate their result. Figure 5C shows the key experiment of this FISH result to validate co-localization of the three detections, but it is difficult to tell if the overlapping-signal is within the nucleus, out of the nucleus or on the surface of the nucleus. Can the authors find a better image?

2) In movie 1, it's difficult to tell that the overlapping-signal is in the nucleus, on the nuclear membrane or outside the nucleus. The Hoechst 33342 channel is too over-saturated to distinguish the boundary of nucleus. The authors need to lower the intensity of Hoechst 33342 channel and may find a better movie. In figure 6 A-C, the Hoechst 33342 channel is also over-saturated to clearly distinguish the boundary of nucleus.

Reviewer #3 (Remarks to the Author):

The revised manuscript has adequately addressed the concerns previously raised by this reviewer. There is only one remaining minor point: in the methods section describing the filters for different fluorescence channels, "emission filter" should be used instead of "emission wheel".

Response to Reviewers' comments:

Reviewer #2 (Remarks to the Author):

The authors have included more details of quantification that improved the quality of the paper. However, there are some problems in the revision that the authors need to further address.

Response: Thank you for the positive comments and suggestions. We have revised our manuscript according to your suggestions.

1) There is a problem in the design of the FISH assay to validate that the TALE-QD co-labeling loci are true targeting site rather than aggregations. The authors used biotin-streptavidin interaction (biotin-HIV DNA probe and Alexa Fluor® 647-streptavidin interaction) for the FISH experiment for validating the TALE-QD signals, but this design is problematic given that the same biotin-streptavidin reaction is already been used to conjugate to TALE-biotin to streptavidin-QDs (TALE N2-R and QD525). Therefore, the FISH result is not valid due to crosstalk between the TALE-N2-R and Alexa Fluor® 647-streptavidin. The authors need to find an orthogonal way to validate their result. Figure 5C shows the key experiment of this FISH result to validate co-localization of the three detections, but it is difficult to tell if the overlapping-signal is within the nucleus, out of the nucleus or on the surface of the nucleus. Can the authors find a better image?

Response: Thank you for your comments and suggestions. The FISH signal could only be acquired by using HIV-specific DNA probes containing many Alexa Fluor® 647-streptavidin binding sites, which were generated from the plasmid pNL4-3 (9.7 kb) labeled with many biotin-dUTPs. In our TALE-QDs labeling system, there is only one TALE-biotin at the target site. Even if the one TALE-biotin bind with Alexa Fluor® 647-streptavidin, the Alexa Fluor® 647 fluorescence could not be detected because of its low brightness. We have carried out a control FISH experiment without biotin-dUTP-labeled DNA probes, and we could not observe the signal of Alexa Fluor® 647 in the TALE-QDs labeling system. Thus, there is actually no crosstalk between the TALE-QDs and the Alexa Fluor® 647 FISH experiments.

According to your suggestion, we have used another image in Figure 5C, which shows more clearly the overlapping-signal within the nucleus.

2) In movie 1, it's difficult to tell that the overlapping-signal is in the nucleus, on the nuclear membrane or outside the nucleus. The Hoechst 33342 channel is too over-saturated to distinguish the boundary of nucleus. The authors need to lower the intensity of Hoechst 33342 channel and may find a better movie. In figure 6 A-C, the Hoechst 33342 channel is also over-saturated to clearly distinguish the boundary of nucleus.

Response: According to your suggestion, we have substituted the movie with a new one in movie 1, and reduced the intensity of Hoechst 33342 channel in figure 6A-C.

Reviewer #3 (Remarks to the Author):

The revised manuscript has adequately addressed the concerns previously raised by this reviewer. There is only one remaining minor point: in the methods section describing the filters for different fluorescence channels, "emission filter" should be used instead of "emission wheel".

Response: Thank you for the positive comments and suggestions. We have changed "emission wheel" to "emission filter" in the revised manuscript.